# Harnessing the Algal Chloroplast for Heterologous Protein Production

**DOI:** 10.3390/microorganisms10040743

**Published:** 2022-03-30

**Authors:** Edoardo Andrea Cutolo, Giulia Mandalà, Luca Dall’Osto, Roberto Bassi

**Affiliations:** Laboratory of Photosynthesis and Bioenergy, Department of Biotechnology, University of Verona, Strada le Grazie 15, 37134 Verona, Italy; edoardoandrea.cutolo@univr.it (E.A.C.); giulia.mandala@univr.it (G.M.); luca.dallosto@univr.it (L.D.)

**Keywords:** microalgae, chloroplast, plastome engineering, recombinant protein production, *Chlamydomonas reinhardtii*, transplastomic biotechnology, molecular pharming, heterologous expression systems, green cell factories, synthetic biology

## Abstract

Photosynthetic microbes are gaining increasing attention as heterologous hosts for the light-driven, low-cost production of high-value recombinant proteins. Recent advances in the manipulation of unicellular algal genomes offer the opportunity to establish engineered strains as safe and viable alternatives to conventional heterotrophic expression systems, including for their use in the feed, food, and biopharmaceutical industries. Due to the relatively small size of their genomes, algal chloroplasts are excellent targets for synthetic biology approaches, and are convenient subcellular sites for the compartmentalized accumulation and storage of products. Different classes of recombinant proteins, including enzymes and peptides with therapeutical applications, have been successfully expressed in the plastid of the model organism *Chlamydomonas reinhardtii*, and of a few other species, highlighting the emerging potential of transplastomic algal biotechnology. In this review, we provide a unified view on the state-of-the-art tools that are available to introduce protein-encoding transgenes in microalgal plastids, and discuss the main (bio)technological bottlenecks that still need to be addressed to develop robust and sustainable green cell biofactories.

## 1. Introduction

The advent of recombinant DNA technology in the 1970s revolutionized biological research and paved the way for modern biotechnology. Starting from the pioneering assembly of artificial circular DNA molecules in vitro [1] and their autonomous replication in a living host [2], genetic engineering achieved its first landmark with the heterologous synthesis of the human peptide hormone somatostatin in a bacterium [3]. The ability to introduce foreign genes in simple biological hosts—typically fast-growing, easily cultivable microorganisms—is paramount to overcome the inherent low yields of the naturally producing species, and to allow the upscaling of production systems.

The last decade has witnessed significant advancements in the genetic engineering of photosynthetic microbes, notably of eukaryotic microalgae of the phylum Chlorophyta, to serve as alternative heterologous hosts [4] to the commonly employed bacteria, yeast, and mammalian cell lines. Several microalgal species are GRAS (generally regarded as safe) organisms and, being oxygenic phototrophs, lead a frugal lifestyle based on light-powered fixation of atmospheric CO_2_, with minimal input requirements compared to heterotrophs. Furthermore, when moved from their natural habitats to controlled (non-nutrient limited) conditions, microalgae display increased growth rates, being suitable for mass culture. Various algal species are amenable to genetic engineering [5], and new strains can be created in a matter of weeks and readily tested for the expression of the products of interest.

Most Chlorophyta contain a large chloroplast—the site of photosynthetic reactions and other metabolic processes—which harbors a compact, tractable genome known as the plastome. Three features render this organelle a preferential target for the expression of recombinant proteins compared to the nucleus: (i) its polyploid condition, (ii) the recombination-mediated targeting of foreign DNA sequences, (iii) the absence of silencing effects.

A prerequisite to develop efficient chloroplast expression strategies is the understanding of the features and workings of this peculiar genetic system. Chloroplast engineering owes much to basic research conducted on plants and on the eukaryotic microalga *Chlamydomonas reinhardtii* [6,7]; today, this species serves as an experimental model with a rich and dedicated genetic engineering toolbox. In this review, we provide a transdisciplinary update on the available strategies to express recombinant products in microalgal plastids (summarized in Figure 1), highlighting the recent advancements in multigenic engineering, inducible expression systems, and biotechnological transfer to non-model strains. We also discuss areas that require major technological innovation and introduce long-term goals that will contribute to developing more robust algal cell factories.

## 2. Chloroplast Genetics in a Nutshell

Known as semi-autonomous organelles, algal chloroplasts are the remnants of a photosynthetic cyanobacteria engulfed by a heterotrophic protist [8]. The bacterial ancestry of plastomes is reflected by the strong AT-rich codon usage bias; therefore, codon optimization is an essential step in the design of chloroplast transgenes [9]. Striking prokaryotic traits of plastids include the transcription of operons in the form of polycistronic units [10], and the bacterial-like composition of the ribosomes [11] causing analogous susceptibility to antibiotics [12,13]. The coevolutionary process between the endosymbiont and the eukaryotic host determined a major horizontal gene transfer to the nucleus [14], resulting in the shrinkage of plastomes to a core set of approximately 80–100 genes. These mostly encode subunits of the photosynthetic apparatus and components of the chloroplast transcription–translation machinery, including the single, plastid-encoded RNA polymerase (PEP) [15]. Nuclear gene expression is thus responsible for the synthesis of most plastid proteins (≈3000) [16], which are imported into the organelle as precursors [17,18].

In *C. reinhardtii*, plastid genes are arranged on a circular chromosome of 204 kb [19] maintained at high polyploidy (≈83 copies) [20], which is uniparentally inherited in sexually reproducing microalgae [21]. Algal plastomes typically display a quadripartite structure, with small and large “single copy” regions divided by two large “inverted repeat” (IR) regions that bear identical gene complements, although the overall genomic architecture can vary significantly between species [22,23]. In *C. reinhardtii*, almost 20% of the plastome consists of non-coding, short dispersed DNA repeats [19]. In addition, to perform structural roles in mRNAs [24], these elements assist in the maintenance of genome integrity by facilitating the conservative homologous recombination repair of double-strand breaks [25] through the activity of a RecA recombinase protein [26]. This same mechanism is exploited during plastome transformation to enable targeted gene replacement between homologous sequences of the artificial DNA and endogenous loci ((1) in Figure 1) [27,28].

During evolution, plastids largely abandoned the prokaryotic regulation of gene expression via the control of transcription initiation [29] in favor of more elaborated post-transcriptional mechanisms [30]. Through such mechanisms, a tight spatiotemporal coordination of local plastid protein synthesis and import is possible during the biogenesis of photosynthetic complexes [31]. To this end, chloroplasts employ an extended network of nucleus-encoded factors that bind in a highly selective way to the 3′- and 5′-untranslated regions (UTRs) of mRNAs [32]. These RNA-binding proteins are crucial components of the anterograde (nucleus-to-chloroplast) system that regulates the expression of plastid genes, being responsible for stabilizing individual transcripts, orchestrating intron splicing, and promoting the maturation of polycistronic pre-mRNAs [33]. In contrast with plants [34], RNA editing does not occur in algal chloroplasts [35]. However, it should be possible to engineer this post-transcriptional regulation of transgene expression in microalgal plastids using synthetic RNA-binding proteins [36]. Hence, the chloroplast represents a unique hybrid genetic system with two prominent eukaryotic features: the existence of both group I and group II introns [37] and their complex trans-splicing [38], and the occurrence of extensive mRNA processing [39]. Finally, the plastid translation machinery is equipped with chaperones [40], such as disulfide- [41] and peptidylprolyl-isomerases [42] and is thus able to support the native folding of complex recombinant proteins.

## 3. Plastome Engineering in a Model Green Alga

Algal plastome transformation was first reported for *C. reinhardtii* in the late 1980s [43,44] using the biolistic technique [45]. Alternative protocols include the agitation of cell wall-deficient (permeable) genotypes in the presence of glass beads [46,47] and electroporation [48]. In its classical design, a chloroplast transgenic cassette must include two external homologous flanking regions enclosing a gene of interest (GOI) and a selectable marker (SM), both equipped with a pair of cis-acting regulatory elements: promoter, upstream 5′- and downstream 3′-UTR ((1) in Figure 1). Codon optimization can be easily performed using online tools, such as the Chlamy Sequence Optimizer [49], while pairs of flanking regions with minimal (≈200 bp) homology [50] are selected ad hoc to introduce transgenes at neutral (non-coding) loci, or to replace disrupted genes. It should be noted, however, that high sequence variability exists between *C. reinhardtii* ecotypes [10], thus the same pairs of homology regions may not work with similar efficiency across different strains. Cassettes of higher complexity can be assembled starting from this minimal scaffold to explore combinations of regulatory elements.

Overall, the ability to perform gene targeting, and the absence of epigenetic silencing, render the plastome a “safe harbor” for transgenes and, thus, the expression of recombinant products. This stands in contrast to nuclear engineering, which usually battles with unpredictable transgene integration, resulting in position effects and methylation-dependent silencing [51]. Although various approaches were pursued to attenuate these constrains [52,53,54,55,56,57], average cytoplasmic yields are still significantly lower compared to chloroplast expression, where the accumulation of the target product can reach up to 21% of total protein content [58].

### 3.1. Selection Strategies

The establishment of transplastomic strains requires, in the first place, reliable selection protocols. The *aadA1* (aminoglycoside adenyltransferase, spectinomycin, streptomycin resistance) [59] and *aphA6* (aminoglycoside 3′-phosphotransferase, kanamycin resistance) [60] antibiotic detoxifying genes are the typical chloroplast selectable markers. Usually, initial transformants are subcultured for several rounds (6–8) on selective media to enrich the transformed chromosome copies until homoplasmy is reached and experimentally verified by PCR. However, it should be noted that a major pitfall of polyploid genomes is the risk of genetic instability, which can arise from persistent heteroplasmy or from spontaneous inter- or intrachromosomal rearrangements ((2) in Figure 1). Accordingly, strategies should be devised to prevent these stochastic events; for example, by including a constant selective pressure in the cultivation strategy. Clearly, antibiotics are not a suitable option in the management of large-scale cultures. Moreover, antibiotic resistance genes should be abandoned in biotechnology because of health and environmental concerns raised over their potential horizontal transfer to other organisms and diffusion through ecosystems [61]. One solution involves SM removal once homoplasmy is established. This is accomplished by engineering short (400–800 bp), homologous, direct repeats flanking the resistance cassette [62] to promote intramolecular recombination and excision upon removal of the selective pressure. Alternatively, fully “antibiotic-free” selection protocols exploit the complementation of recipient strains affected in essential metabolic activities, usually acetate-requiring strains with lesions in plastid photosynthetic genes. Photosynthesis-competent revertants can be selected following transformation with a functional copy of the missing/disrupted gene and selection of minimal medium. Commonly used strains have deletions in the following loci: *atpB* (beta subunit of the ATPase complex) [63], *psbA* (D1 protein of photosystem II, PSII) [64], *rbcL* (RuBisCO large subunit [65], *petB* (cytochrome 6b) [66], *tscA* (trans-splicing factor) [67,68], and *psbH* (subunit H of PSII) [69]. A recent version of the latter was created by disrupting the *psbH* locus with the *aadA1* cassette in the cell wall-deficient strain cw15 (TN72, CC-5168) [47,70]. Transformants are obtained via glass agitation by retrofitting the functional *psbH* gene and, when homoplasmic, are sensitive to spectinomycin due to the loss of the *aadA1* cassette. However, despite their widespread use, non-photosynthetic mutants have two main drawbacks: (i) the locus-specific targeting requirement with an ad hoc recipient strain, (ii) the need to include the complementing gene in the transformation cassette.

A recent innovation in algal biotechnology is the adoption of the *ptxD* gene, encoding the NAD^+^-dependent phosphite oxidoreductase enzyme from the bacterium *Pseudomonas stutzeri* [71]. PTXD catalyzes the conversion of phosphite ions (Phi, PO_3_^3−^) into phosphate (Pi, PO_4_^3−^), making this unharmful, reduced phosphorus compound a very convenient selective agent ((3) in Figure 1). In fact, most organisms—including the parasites that commonly infest microalgal cultures—cannot readily assimilate Phi in their metabolism, while PTXD-expressing transgenics can grow when Phi is provided as the sole source of the essential nutrient phosphorus. Accordingly, *ptxD* was first introduced into the nucleus of *C. reinhardtii* [72,73] to develop a Phi-based pest control strategy to enable axenic microalgal cultivation in non-sterile conditions. Chloroplast *ptxD* expression was subsequently reported in different studies [74,75,76] and, recently, an engineered enzyme isoform, more suited for the chloroplast biochemical environment, was developed. This catalytically flexible PTXD version uses NAD^+^ (limiting in the chloroplast) and NADP^+^ (not limiting in the chloroplast) nicotinamide cofactors with equal efficiency and could be reliably employed as a metabolic selectable marker to generate plastid transformants via the direct selection on Phi, both in *C. reinhardtii* [77] and in the non-model species *Picochlorum* [78]. In a proof-of-principle approach, PTXD was translationally fused to a GOI, creating an in vivo cleavable protein chimera [79]. In this instance, fertilization with Phi imposes a strict metabolic pressure against the risk of genetic instability, ensuring transgene(s) maintenance in the plastome, and affording the safe upscalability of cultivation systems with a concomitant reduction in management costs.

### 3.2. Cis-Acting Regulatory Elements

Efficient heterologous expression requires the association of cis-acting elements to transgenes to regulate their transcription and translation. In addition to promoters, 5′- and 3′-UTRs are critical features of chloroplast engineering [80], since these sequences are known to play crucial structural roles [81,82,83] and influence the stability [84,85,86], lifetime [87,88], and translation [89,90,91] of endogenous mRNAs. Accordingly, the expression of recombinant products in the chloroplast is significantly influenced by the choice of these regulatory sequences [92]. Traditionally, cis-acting elements derived from photosynthetic genes (*atpA*, *psaA*, *rbcL*, *psbD,* and *psbA*), or *16S* RNA, are employed to regulate transgene expression [93]. Several efforts were made to define the most efficient regulatory sequences, including the hybrid combinations thereof [90,94,95,96].

The *psbA* promoter/5′-UTR pair deserves a special mention. This strong, light-inducible element is known to be repressed by the synthesis of its product through a negative feedback loop acting on the *psbA* mRNA 5′-UTR [97]. Therefore, when transgenes are placed under the transcriptional control of the *psbA* promoter, they will likely suffer from this endogenous attenuation, resulting in poor expression. Accordingly, the *psbA* promoter is commonly used in non-photosynthetic *psbA*-deficient backgrounds via heterotrophic cultivation, thus restricting the light-driven production of recombinant molecules. A first solution to this issue involved the use the *psbD* promoter to complement *psbA* transcription (Manuell et al., 2007) and restore photosynthetic competence while exploiting the *psbA* element to drive efficient transgene expression. More recently, two studies employed *psbA* genes from other photosynthetic organisms [98], including interspecific *psbA* promoter/5′-UTR combinations [99], to relieve the interference on transgenes. These experiments clearly showed that it is possible to introduce completely heterologous regulatory elements to drive transgene expression in microalgae. This principle was recently extended to a complete foreign plant system, where nucleus-encoded trans-acting factors from *Arabidopsis thaliana* and *Zea mays* were used in *C. reinhardtii* to stabilize plastid transgenes bearing the native plant recognition sequence on their 5′-UTR [100].

Altogether, it appears that cis elements can result in high expression variability depending on the associated transgene [86,95,101]; thus, empirical optimization is required. Today, modular cloning techniques enable the seamless fusion of elements, and explore combinatorial cassette design. In this respect, the MoClo (modular cloning) toolkit [102] significantly facilitated nuclear engineering efforts in *C. reinhardtii*. However, despite earlier attempts [64,103], a comprehensive library of validated standard genetic elements and cloning system for algal plastomes is still missing, and is expected to drastically accelerate innovation in this field.

### 3.3. Reporter Genes

The efforts to improve recombinant protein yields can significantly benefit from reporter systems to non-invasively monitor transgene expression in vivo [9,104]. The green fluorescent protein (GFP) and its derivatives are particularly useful tools since they enable the rapid screening of transformants via microscopy- or flow cytometry-based techniques. Recent innovations in this respect include chloroplast-derived optimized isoforms of the vivid Verde (VFP, green) [105] and mCherry (red) fluorescent proteins [106]. These reporters can also be employed to assess the impact of different cultivation parameters—mainly light regime, temperature, and metabolism—on recombinant protein expression. For instance, a recent study reported that mixotrophic growth (a combination of heterotrophic acetate fermentative consumption and photoautotrophy) with moderately low light (35 μE m^−2^ s^−1^) resulted in the highest accumulation of a GFP-fused bacterial lytic protein under the control of the *16S* promoter [107]. A novel, high-throughput, fluorescence-based cell sorting system was recently described and validated to select GFP-expressing nuclear transformants of two non-model algal species [108]. This method enables the concomitant assessment of photosynthetic parameters, and is expected to reveal useful applications in transplastomic technology.

**Figure 1 microorganisms-10-00743-f001:**
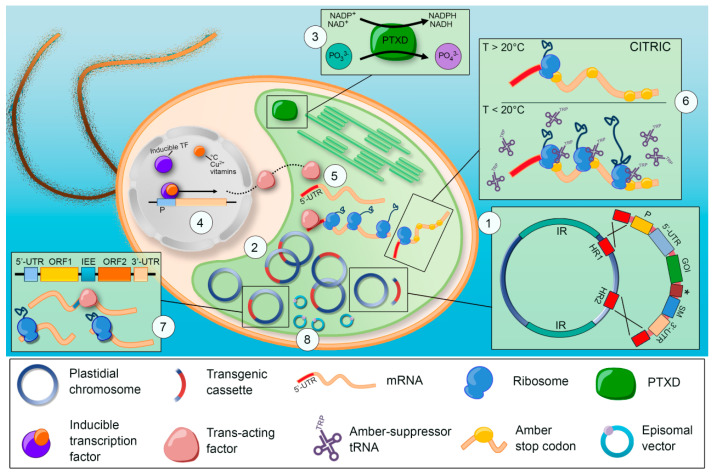
Graphical summary of emerging tools for chloroplast engineering in microalgae. (**1**) Typical quadripartite structure of a plastid chromosome with two inverted repeats (IRs) and transgene integration enabled by recombination between homology regions (HR1-2). The gene(s) of interest (GOI) and the selectable marker (SM) are connected by a linking element (*), or individually equipped with cis-regulatory sequences: promoter (P), 5′- and 3′-UTRs (untranslated regions). (**2**) Under constant selection, the plastome is enriched in transformed chromosomes (blue–red circles), although untransformed copies (blue circle) may persist and expose the system to the risk of genetic instability. In addition, spontaneous inter- or intrachromosomal recombination events may lead to transgene loss. (**3**) The PTXD enzyme performs the conversion of phosphite ions (PO_3_^3−^) into phosphate (PO_4_^3−^) and serves as a metabolic selectable marker, also enabling axenic algal cultivation in non-sterile media [77]. (**4**) Transgene expression can be finely controlled via chemical- or temperature-inducible, nucleus-encoded, trans-acting factors that (**5**) bind the 5′-UTRs of plastid mRNAs and promote their translation [109]. (**6**) The CITRIC (cold-inducible translational readthrough in chloroplasts) system requires plastome manipulation only and exploits a temperature-sensitive suppressor tRNA to regulate translation [110]. (**7**) Polycistronic constructs, in which multiple open reading frames (ORFs) are connected via native intercistronic expression elements (IEEs), are processed by endogenous trans-acting factors into separate mono-cistronic transcripts that are independently translated [111]. (**8**) As pioneered in plant plastids, transgene expression can potentially be achieved in microalgae via episomal vectors that do not require integration into the circular chromosome, but are stably maintained by the host.

## 4. Examples of Recombinant Products

The list of recombinant products expressed in the chloroplast of the model species *C. reinhardtii* embraces several categories, truly testifying the versatility of this heterologous system. A comprehensive list is reported in Table 1, along with the used selection strategy, cis-regulatory elements, and other relevant information for each work.

A major advantage of microalgae is that the whole harvested biomass contains the product(s) of interest. Following processing, products can be easily retrieved via peptide tags [112,113]. In this respect, cell wall-deficient mutants are particularly useful, since their cells are easily disrupted with mild forces [114]; however, this feature also makes them susceptible to the shear forces present during biomass collection [115]. Alternatively, given the non-toxic status [116] and digestibility of most algal species [117], the biomass can be processed into edible lyophilized capsules that survive the gastric environment, preserving the biological activity of the products [118].

### 4.1. Microalgal Molecular Pharming

The possibility to exploit the light-driven conversion of CO_2_ into high-value heterologous molecules makes the algal chloroplast an excellent system for the low-cost production of biopharmaceuticals [119,120]. However, it should be noted that glycosylation, an essential post-translational modification process required for the biological activity of many therapeutical proteins [121], cannot be achieved in the chloroplast [122], thus limiting the range of potential products. Nevertheless, successful chloroplast expression was reported for antimicrobial peptides [99,107,123,124,125] and human proteins, with applications in regenerative medicine [70,126,127,128] and the treatment of hypertension [129,130]. Notably, algal chloroplasts can assemble full-length human immunoglobulins [131], including monoclonal antibodies against toxins [132,133,134,135], whose sequestration in the organelle is beneficial to prevent detrimental effects on eukaryotic cell components.

Arguably, the most promising biomedical application of microalgae is the development of orally deliverable subunit vaccines and other immunizing agents [136], such as allergens [137]. Examples include epitopes of the foot-and-mouth disease virus [138], the bacterium *Staphylococcus aureus* [139], and diabetes- [140] and atherosclerosis-associated autoantigens [141]. A series of works, instead, produced surface epitopes of the malaria-causing agent *Plasmodium falciparum* [112,113,142,143,144,145,146] that could elicit transmission-blocking antibodies in mice. Importantly, a major constraint of expressing viral antigens in algal chloroplasts is their susceptibility to unspecific proteolytic degradation [58]. This likely explains why the race to develop a vaccine against the SARS-CoV-2 virus has not yet produced a chloroplast-based candidate. Recently, cytoplasmic accumulation of an epitope corresponding to the viral receptor-binding domain (RBD) was achieved following the nuclear transformation of *C. reinhardtii*. However, when the polypeptide was targeted to the chloroplast, a significant portion of the RBD moiety was lost [147].

In addition to human health, transplastomic microalgae are finding emerging applications in aquaculture [148]. Few studies reported the production antigens of fish- [95] and shrimp-associated pathogens [58,149] that efficiently prevent infections. Alongside antigenic peptides, algal chloroplasts are also suitable expression systems from which to manufacture double-stranded RNA targeting common fish viruses [150]. Reported veterinary applications of transplastomic microalgae include the production of swine-associated pathogens [151] and/or enzyme-based dietary supplements for aviculture [152].

### 4.2. Enzymes, Metabolites, and Valorization of Lignocellulose

The algal chloroplast has been proposed as a biofactory from which to produce heterologous enzymes with industrial applications [65], and for biomanufacturing non-native, high-value metabolites. Indeed, the chloroplast harbors an extended metabolic network, into which new enzymes or pathways can be introduced to produce synthetic metabolites. Metabolic engineering in algal plastids is still in its infancy; however, the potential of this organelle to serve as an experimental system for synthetic biology approaches has already been acknowledged [153]. Indeed, the plastid contains several biosynthetic precursors that can be used to produce high-value metabolites. For instance, an early study on *C. reinhardtii* showed that chloroplast expression of the β-carotene hydroxylase gene (*crtR-B*) from *Haematococcus pluvialis* (*lacustris*) resulted in the overaccumulation of carotenoids, including the potent antioxidant pigment astaxanthin [154]. Recently, the whole astaxanthin pathway operon was introduced into transplastomic tobacco (*Nicotiana tabacum*) plants, demonstrating the feasibility of complex chloroplast metabolic engineering to produce this heterologous metabolite [155]. Isoprenoids (or terpenes) are another class of metabolites that could be heterologously manufactured in microalgae [156]. These molecules have several applications in human health [157], and can be produced in plastids, starting from the precursors originating from the endogenous carotenoid biosynthetic pathways. In this respect, two studies have successfully expressed a bifunctional diterpene synthase [158], and a plant cytochrome P450 [159], in the chloroplast of *C. reinhardtii*, showing that these enzymes do not interfere with algal physiology in pilot-scale cultivation [160].

Another emerging application of transplastomic microalgae is their use in the production of renewable energy through the saccharification of lignocellulose. Initially attempted in the halophytic species *Dunaliella tertiolecta* [161], this strategy was subsequently implemented in *C. reinhardtii* [162,163] with the expression of hyperthermophilic cellulases [79,164]. In this application, the dried algal powders containing the recombinant enzymes are mixed at high temperatures with the raw biomass to promote the release of fermentable sugars. To achieve full valorization of lignocellulose, however, additional enzymes with auxiliary activities are required to promote the hydrolysis of recalcitrant components. In this respect, lytic polysaccharide monooxygenases [165] and lignin-modifying enzymes [166] are suitable candidates. These are typically cuproenzymes (copper-requiring), whose production should not be limited by the availability of this element in the chloroplast [167].

**Table 1 microorganisms-10-00743-t001:** Recombinant products expressed in the chloroplast of the model alga *C. reinhardtii*.

Expressed Product	Category/Application	Promoters and Cis-Acting Elements	Selection System	Highlights	References
**Biopharmaceuticals/green biologics for human health**					
Bovine mammary-associated serum amyloid protein (M-SAA)	Prophylaxis of bacterial and viral infections	P*psbA* *psbA* 3′-UTR	*aadA1* gene (aminoglycoside adenyltransferase)—Spectinomycin	Accumulation of 5% of TSP	
Bioactive peptides from milk proteins connected	Antihypertensive, antimicrobial, immunomodulatory, antioxidant, and hypocholesterolemic activities	P*rbcL* or P*atpA* *rbcL* 3′-UTR	*aadA1* gene—Spectinomycin	Chimeric peptides linked by gastrointestinal proteases cleavage sites	[124]
Bovine milk amyloid A protein (mammary-associated serum amyloid A, M-SAA)	Prophylaxis of bacterial and viral infections	Combinations of various endogenous and heterologous promoters (*psbA, atpA, tufA*, and *psbD*)	Phototrophic rescue of Δ*psbA* strain and *aphA6* gene (aminoglycoside 3′-phosphotransferase)—Kanamycin	Avoidance of *psbA* auto-attenuation and photoautotrophic growth in 100 L	[99]
Tenth human fibronectin type III domain (10FN3)	Extracellular matrix glycoprotein with roles in cell adhesion, migration, growth, and differentiation	P*psbA* *psbA* 3′-UTR	*aphA6* gene—Kanamycin	A carboxy-terminal fusion to the M-SAA protein enabled synthesis of otherwise non detectable products	[126,127]
Fourteenth human fibronectin type III domain (14FN3)	Antibody mimic				
Human vascular endothelial growth factor (VEGF) isoform 121	Treatment of pulmonary emphysema				
High mobility group protein B1 (HMGB1)	Mediator of wound healing				
Human growth hormone (hGH)	Growth hormone deficiency	P*psaA* and P*atpA rbcL* 3′-UTR	Phototrophic rescue of TN72 (Δ*psbH*) strain	The purified hGH has biological activity in vitro	[70]
Chimeric antihypertensive peptides (angiotensin-converting enzyme ACE-inhibitory peptides)	Treatment of hypertension	P*rbcL* *rbcL* 3′-UTR	*aadA1* gene—Spectinomycin	Antihypertensive and ACE-inhibitory effects of the recombinant protein demonstrated in vivo in murine models	[129]
Chimeric antihypertensive peptides (angiotensin-converting enzyme ACE-inhibitory peptides)	Treatment of hypertension	P*rbcL* *rbcL* 3′-UTR	*aadA1* gene—Spectinomycin	Antihypertensive effect of the recombinant protein demonstrated in vivo in murine models	[130]
Phosphorylated human osteopontin	Bone regenerative therapy	Not disclosed	Not disclosed	Successful specific folding and PTMs	[128]
Full-length IgG1 human monoclonal antibody against anthrax protective antigen 83 (PA83) (heavy and light chains, HC, LC)	Blocker of anthrax toxin	P*psbA* *rbcL* 3′-UTR (HC) P*psbA* *psbA* 3′-UTR (LC)	Mutated *1**6S*-*rRNA* gene—Spectinomycin	The antibody binds its target antigen, PA83, at levels similar to the same antibody expressed in mammalian cells	[131]
Variable domains of camelid heavy chain-only antibodies (VH H) binding and neutralizing botulinum neurotoxin	Antitoxin	P*psbA* *psbA* 3′-UTR	*aphA6* gene—Kanamycin	Accumulation of 5% of TSP Prevention of neuron intoxication in vitro Stable in gastric environment	[134]
Single-chain fragment variable (scFv) antibody	Proof-of-concept production of bioactive recombinant protein	P*psaA* *rbcL* 3′-UTR	Phototrophic rescue of TN72 (Δ*psbH*) strain	Fusion to the Tat export signal peptide-enabled targeting of the recombinant product in the thylakoid lumen	[135]
Bacteriophage Cpl-1 and Pal endolysins	Antibacterial effectors against *Streptococcus pneumoniae*	P*psaA*-exon 1 *rbcL* 3′-UTR	Phototrophic rescue of TN72 (Δ*psbH*) strain	Demonstrated antibacterial activity against different serotypes of *S. pneumoniae,* including clinical isolates	[125]
PlyGBS bacterial lysin	Antibacterial effector against *Streptococcus*	P*16S* rRNA *atpA* 5′- and 3′-UTRs	*aadA1* gene—Spectinomycin	The effect of light period and intensity on recombinant protein expression was investigated, revealing optimal conditions with mixotrophy under constant illumination at moderately low light (35 μE m^−2^ s^−1^)	[107]
Single-chain antibody (scFv) targeting the B-cell surface antigen CD22 fused to the enzymatic domain of exotoxin A from *Pseudomonas aeruginosa* (immunotoxin)	Treatment of B-cell lymphomas	P*psbA* *psbA* 3′-UTR	*aphA6* gene—Kanamycin	The expressed proteins specifically bind and reduce the viability of B-cell lymphomas in vitro	[133]
Single-chain antibody (scFv) targeting the B-cell surface antigen CD22 fused to the eukaryotic ribosome inactivating protein, gelonin (immunotoxin)	Treatment of B-cell lymphomas	P*psbA* *psbA* 3′-UTR	*aphA6* gene—Kanamycin	The expressed proteins specifically bind and reduce the viability of B-cell lymphomas in vitro	[132]
Major birch pollen allergen Bet v 1	Allergen immunotherapy (AIT) for the the treatment of allergic diseases	P*psaA* *rbcL* 3′-UTR	Phototrophic rescue of FUD50 (Δ*atpB*) strain	The Bet v 1 antigen from algae showed similar binding to human IgE and murine Bet v 1-specific IgG	[137]
Foot-and-mouth disease virus VP1 protein fused with cholera toxin B	Antigen Adjuvant	P*atpA* *rbcL* 3′-UTR	*aadA1* gene—Spectinomycin	The fusion protein displayed GM1-ganglioside-binding affinity and antigenicity	[138]
Human glutamic acid decarboxylase (hGAD65) (diabetes-associated autoantigen)	Diagnostic marker/antigen for immunotherapy	P*rbcL* *rbcL* 3′-UTR	*aadA1* gene—Spectinomycin	Antigenicity of algal derived product verified by ELISA and in vivo assays	[140]
D2 fibronectin-binding domain of *Staphylococcus aureus* fused with the cholera toxin B subunit (CTB)	Oral vaccine	P*rbcL* *rbcL* 3′-UTR	*aadA1* gene—Spectinomycin	Induction of specific mucosal and systemic immune responses in mice	[139]
p210 epitope from apolipoprotein ApoB100 fused to the β subunit of the cholera toxin (CtxB)	Oral vaccine/immunotherapy for atherosclerosis	P*atpA* *rbcL* 3′-UTR	*aadA1* gene—Spectinomycin	In vivo immunogenic activity of the chimera when orally administered in mice and detection of anti-p210 serum antibodies	[141]
*Plasmodium falciparum* surface protein 25 (Pfs25) and 28 (Pfs28)	Malaria antigen/subunit vaccines	*PpsbA* *psbA* 3′-UTRs	*aphA6* gene—Kanamycin	The two proteins are immunogenic in mice and Pfs25 antibodies bind in vitro to *P. falciparum*, exhibiting transmission-blocking activity	[142]
C-terminal antigenic domain of the *Plasmodium falciparum* surface protein Pfs48 and 45	Malaria antigen/transmission-blocking vaccine	P*psbD* *psbA*-3′-UTR P*psbA* *psbA*-3′-UTR	Phototrophic rescue of *psbH*—strain	The purified peptides are recognized by specific transmission-blocking antibodies	[112]
*Plasmodium falciparum* surface protein 25 (Pfs25) fused to the β subunit of the cholera toxin (CtxB)	Malaria antigen/transmission-blocking vaccine	P*psbA* *psbA* 3′-UTR	*aphA6* gene—Kanamycin	Orally vaccinated mice with freeze-dried algae containing CtxB-Pfs25-elicited CtxB-specific serum IgG antibodies, and both CtxB- and Pfs25-specific secretory IgA antibodies	[143,144]
Single-chain fragment antibody molecule (αCD22 scFv) and *Plasmodium falciparum* surface protein 25 (Pfs25)	Malaria antigen/transmission-blocking vaccine	P*psbA* *psbA* 3′-UTR	*aphA6* gene—Kanamycin	Optimization of light intensity (300 μmol m^−2^ s^−1^) resulted in six-fold increase in protein accumulation	[113]
*Plasmodium falciparum* PfCelTOS antigen (cell traversal protein for ookinetes and sporozoites) alone and fused to human interleukin-2 (IL-2)	Malaria antigen/transmission-blocking vaccine fused to adjuvant	P*atpA* *rbcL* 3′-UTR	Phototrophic rescue of TN72 (Δ*psbH*) strain	Protein accumulation is promoted by mixotrophic cultivation in low light	[145,146]
**Biopharmaceuticals and enzymes for aquaculture, animal health, and pest control**					
*vapA* and *acrV* proteins from the fish pathogen *Aeromonas salmonicida*	Antigen/immunization	Various combinations of promoters, 5′- and 3′-UTRs	Phototrophic rescue of FUD50 (Δ*atpB*) strain *aadA1* gene—Spectinomycin	Strongest expression with the P*psaA*-exon1/5′-UTR element	[95]
VP28 protein of the white spot syndrome virus	Oral vaccine	P*psbA* *psbA* 3′-UTR	*aadA1* gene—Spectinomycin	Recombinant product accumulated to 21% of TCP	[58]
Classical swine fever virus (CSFV) structural protein	Antigen/immunization	P*atpA* *rbcL* 3′-UTR	*aadA1* gene—Spectinomycin	Accumulation of 1.5–1% of TSP, antigenicity verified by ELISA	[151]
*Escherichia coli* AppA phytase enzyme	Feed additive for poultry	P*atpA* *rbcL* 3′-UTR	Mutated *1**6S*-*rRNA* gene—Spectinomycin	Fecal phytate excretion is reduced following feeding with whole-cell algal lysate	[152]
Cry (1A, 4A, 4B and 11A) cytotoxic proteins of *Bacillus thuringiensis* subsp. *israelensis* (Bti)	Mosquito control	*PpsbD* *psbA* 3′-UTR	Mutated *1**6S*-*rRNA* gene—Spectinomycin	Live cell bioassays demonstrated toxicity of the *cry* transformants to larvae of *Aedes aegypti* and *Culex quinquefasciatus*	[168,169]
VP28 protein of the white spot syndrome virus (WSSV)	Oral delivery system to control WSSV disease in shrimp	P*atpA* *rbcL* 3′-UTR	Phototrophic rescue of TN72 (Δ*psbH*) strain	Feeding of algal biomass exressing the VP28 antigen improved shrimp survival upon infection with WSSV	[149]
**Enzymes with industrial applications**					
Alcohol dehydrogenase (ADH1) from *Saccharomyces cerevisiae*	Ethanol production	P*rbcL* *rbcL* 3′-UTR	Phototrophic rescue of *rbcL* (CC2653) mutant	Algal cultivation in low oxygen partial pressure or anoxia promoted ADH1 accumulation and ethanol production	[65]
β-carotene hydroxylase (crtR-B) from *Haematococcus pluvialis (lacustris)*	Metabolic engineering/heterologous synthesis of astaxanthin	P*atpA* *rbcL* 3′-UTR	*aadA1* geneSpectinomycin	Total carotenoid content is increased in the crtR-B transformants following high light treatment compared to wild type cells	[154]
Plant cytochrome P450 (CYP79A1)	Metabolic engineering/heterologous synthesis of diterpenoids	P*atpA* *rbcL* 3′-UTR	Phototrophic rescue of TN72 (Δ*psbH*) strain	The enzyme is targeted to the organelle membrane via its endogenous N-terminal region, and converts tyrosine to *p*-hydroxyphenylacetaldoxime	[159]
Bifunctional diterpene synthase (*cis*-abienol synthase, TPS4)	Metabolic engineering/heterologous synthesis of diterpenoids	P*atpA* *rbcL* 3′-UTR	Phototrophic rescue of TN72 (Δ*psbH*) strain	The expression of this enzyme is compatible with pilot-scale algal cultivation [160]	[158]
Glycohydrolase family 5 endoglucanase from *Paenibacillus* sp. KCTC8848P (CelK1)	Hydrolytic enzyme/saccharification of lignocellulosic biomass	P*psaA* *rbcL* 3′-UTR	Phototrophic rescue of FUD50 (Δ*atpB*) strain		[162]
*Cel6A* endoglucanase from *Thermobifida fusca*	Hydrolytic enzyme/saccharification of lignocellulosic biomass	P*16S rRNA* *atpA* 5′-UTR *rbcL* 3′-UTR	*aadA1* gene—Spectinomycin	A fusion to the downstream box (DB) of the *TetC* (tetracycline cyclase) gene improved protein accumulation	[163]
CelB endoglucanase from *Thermotoga neapolitana*	Hyperthermophilic hydrolytic enzyme/saccharification of lignocellulosic biomass	P*psaA* *rbcL* 3′-UTR	*aadA1* gene—Spectinomycin	The cellulolytic blend enabled the conversion of alkaline-treated lignocellulose into glucose Hydrolysates boosted the biogas production by methanogenic bacteria	[164]
Cellobiohydrolase portion of the CelB cellulosome (C-CBH) from *Caldicellulosiruptor saccharolyticus*					
β-glucosidase from *Pyrococcus furiosus*					
Xylanase from *Thermotoga neapolitana*					
CelB endoglucanase from *Thermotoga neapolitana*	Hyperthermophilic hydrolytic enzyme/saccharification of lignocellulosic biomass	P*psaA* *rbcL* 3′-UTR	Phototrophic rescue of *FUD50* (Δ*atpB*) strain	Expression of the hydrolytic enzyme is coupled to the PTXD growth selector, enabling selective growth in non-sterile, phosphite-fertilized medium	[79]

P, promoter; UTR, untranslated region.

## 5. Multigenic Engineering

Full exploitation of transplastomic microalgae requires the ability to introduce multiple open reading frames (ORFs) to implement metabolic pathways and/or enable the synthesis of multi-subunit proteins. To date, microalgal plastome engineering has mostly relied on the insertion of a single transgene coupled to a SM, while multigenic strategies are already established in several plant species [155,170,171,172].

A relatively simple strategy attempted in microalgae involves the use of endogenous Shine–Dalgarno-like sequences to connect independent ORFs exploiting the prokaryotic mechanism of ribosome reinitiation [124,129,141,173]. Another promising approach is based on polycistronic transgenic units that reflect the endogenous organization and expression of gene batteries in plastids [10,39]. This strategy was pioneered in plants [171,174,175] and was only recently attempted in microalgal plastids [111]. In this system, short intergenic sequences, known as intercistronic expression elements (IEEs), are used to connect multiple ORFs in transgenic operons ((7) in Figure 1). IEEs are recognized by nucleus-encoded trans-acting factors that assist the maturation of the polycistronic pre-mRNAs, enabling the independent translation of cistrons [33]. This principle was exploited to express a bicistronic construct composed of a SM and a GFP reporter using two IEEs derived from the endogenous *psbN-psbH* and *tscA-chlN* gene pairs [111].

Alternative strategies are based on the simultaneous delivery of multiple ORFs, either via single [176] or sequential transformation events [177]. It should be noted, however, that the sequential approach is limited by the availability of SMs and cis-regulatory elements. In addition, the repetitive use of genetic elements in the transgenic cassettes may favor spontaneous intramolecular recombination and genetic instability. This issue was highlighted in a recent study [177], where the association of the *rbcL* 3′-UTR element in three adjacent ORFs resulted in the loss of two genes, although this unwanted outcome could be prevented by using a shorter version of this sequence.

An earlier synthetic biology approach demonstrated the possibility of performing multigenic engineering using, as a testbed, a series of endogenous photosynthetic genes [176]. Initially, the authors sequentially removed six unrelated genes, and then reintroduced them in the form of single transgenic cluster, where each ORF was equipped with its native cis-regulatory elements; interestingly, the complemented strains displayed suboptimal photosynthetic efficiency compared to the parental genotype. Overall, the task of chloroplast multigenic engineering in *C. reinhardtii* is open for major improvements.

## 6. Inducible Expression Systems

A desirable feature of heterologous systems is the ability to temporally control transgene expression to avoid potentially toxic effects of foreign peptides (or metabolites) on the host’s physiology. In addition, the constitutive hyperaccumulation of recombinant products in the chloroplast can override the quality control machinery and activate the unfolded protein response, leading to their proteolytic degradation [178,179].

An early type of inducible system exploited the nucleus-encoded maturation factor Nac2, a trans-acting element belonging to the tetratricopeptide-like protein family that natively stabilizes the 5′-UTR of the plastid *psbD* transcript [180,181] to regulate the expression of chloroplast transgenes ((4,5) in Figure 1). By placing *Nac2* under the transcriptional control of copper- [109,168,182] or vitamin B12-repressible promoters [183,184], or via fusion to a thiamine pyrophosphate riboswitch [185,186], it is possible to tightly regulate the translation of plastid transgenes equipped with the *psbD* 5′-UTR.

In addition to chemical inducers, temperature-shifts can be used to control transgene expression [187]. Initially, the hybrid *Hsp70A-Rbcs2* promoter, which is activated above 40 °C, was used to regulate the expression of the nuclear-encoded protein TDA1 (chloroplast translation factor 1), which in turn stabilizes transgenes bearing its native binding site, the *atpA* 5′-UTR [188,189].

A recently developed temperature-controlled expression system named CITRIC (cold-inducible translational readthrough in chloroplasts), instead, requires only plastome engineering [110] ((6) in Figure 1). This approach is based on two elements: (i) an amber (UAG) suppressor tRNA gene [190], which is only stable below 20 °C [191], and (ii) a GOI-harboring multiple UGA within its coding sequence. Notably, none of the plastid genes contains the amber codon [192], although it is recognized as a terminator. In this system, above 20 °C, the nascent recombinant polypeptide undergoes premature translation abortion, while at lower temperature the engineered tRNA enables its full synthesis. This codon reassignment principle has two additional benefits. On the one hand, it prevents potential toxic effects of leaky chloroplast promoters during cloning operations in bacteria [193]. On the other hand, this strategy can be exploited for biocontainment purposes to reduce the risk of transgene flow and/or escape from genetically engineered microalgae [76,194].

## 7. Beyond Model Species

A crucial step for the advancement of algal biotechnology is the transition towards non-model, more robust algal species. The phylum Chlorophyta is truly an untapped reservoir of biodiversity, comprising over 4500 species [195]. Many evolved in extreme ecological niches, and possess physiological characteristics that are relevant for biotechnological exploitation. However, because of recalcitrance to transformation and/or a lack of customized tools, only a few microalgal species have been genetically engineered so far [196]. As previously reported, plastid promoters and UTRs from different photosynthetic organisms are interchangeable between species [98]; however, it is always preferable to use native sequences to drive transgene expression. In this respect, genomic resources are essential tools to enable the customization of cassettes and biotechnological transfer. To this end, the online repository OGDA (organelle genome database for algae, http://ogda.ytu.edu.cn/ accesses on 27 March 2022) [197] enables the visualization of sequenced algal plastomes, while the ChloroMitoCU resource facilitates the selection of synonymous codons for each species [198].

Of particular interest for biotechnology are microalgae that withstand environmental fluctuations, and with fast generation times. Accordingly, bioprospecting endeavors have focused on resilient species amenable to outdoor cultivation [199]. Reports of chloroplast transformation of non-model species are listed in Table 2, along with information regarding the insertion loci, regulatory elements, selection strategies, and expressed products. These include halophilic organisms, such as *Tetraselmis subcordiformis* [200,201]; two subspecies of the β-carotene-accumulating *Dunaliella* genus *D. salina* [202] and *D. tertiolecta* [161]; *Desmodesmus armatus* [203]; and the biological producer of the potent antioxidant pigment astaxanthin, *H. pluvialis* (lacustris) [204,205,206].

Of all Chlorophyta, species belonging to the class Trebouxiophyceae are regarded as the most promising candidates for industrial applications due to their thermotolerance, resistance to extreme irradiance, and high growth rates. For instance, the species *Chlorella sorokiniana* can be cultivated at very high density under heterotrophic conditions, with reported fresh biomass production of 250 g L^−1^ in 1000 L fermenters [207], while its related species *C. ohadii* thrives at extremely high light intensities due to its exceptionally efficient photoprotective mechanisms [208]. However, despite the suitability of these species for large-scale outdoor cultivation, and the availability of sequenced plastomes [209], their exploitation in the production of recombinant molecules is currently hampered by recalcitrance to genetic transformation, mostly due the presence of a thick cell wall [210]. So far, successful chloroplast transformation was reported in three Trebouxiophyceae species: *C. vulgaris* [211], a species lacking IRs in its plastome [212]; *Parachlorella kessleri* [213]; and the emerging halophilic species, *Picochlorum renovo* [78,173]. Chloroplast engineering was also described for stramenopiles, including the oleaginous alga *Nannochloropsis oceanica* [48] and the closely related species *N. gaditiana* [214], and the diatom *Phaeodactylum tricornutum* [215]. Stramenopiles are currently regarded as promising green biofactories due to their stress-tolerant and high lipid-accumulating phenotype, and their amenability to genetic manipulation due to an expanding genetic engineering toolbox [216].

Overall, the combined efforts of bioprospecting and genome sequencing, such as in the recent case of the high-light-tolerant chlorophyte, *Asterarcys* sp. [217], will favor the domestication [218] of industrially relevant microalgal species, leading to more efficient transplastomic expression platforms.

**Table 2 microorganisms-10-00743-t002:** Reported chloroplast transformation in non-model microalgae.

Species	Class, Order, and Family	Physiological Characteristics	Transformation Method and Selection System	Plastome Integration Site(s)	Expressed Product and Cis-Regulatory Elements	Reference
**Green algae (Chlorophyta)**						
*Tetraselmis subcordiformis*	Chlorodendrophyceace, Chlorodendrales, Chlorodendraceae	Halophilic	Biolistics *Bar* gene (phosphinothricin N-acetyltransferase)—Bialaphos	Silent site between *rrn16S/tRNA-I* and *tRNA-A/rrn23S*	Enhanced green fluorescent protein (eGFP) Endogenous P*atpA* *C. reinhardtii rbcL* 3′-UTR	[200]
*Tetraselmis subcordiformis*	Chlorodendrophyceace, Chlorodendrales, Chlorodendraceae	Halophilic	Biolistics *Bar* gene—Bialaphos	Silent site between *rrn16S/tRNA-I* and *tRNA-A/rrn23S*	Enhanced green fluorescent protein (eGFP) Combination of endogenous elements from *rrn16S*, *rbcL*, *psbA*, and *psbC* *Prrn*-*psbA*-5′UTR–*psb*A 3′-UTR produced highest expression	[201]
*Dunaliella salina*	Chlorophyceae, Chlamydomonadales, Dunaliellaceae	Halophilic	Biolistics *hptll* gene (hygromycin phosphotransferase)—Hygromycin	Silent site between *rrn16S/tRNA-I* and *tRNA-A/rrn23S*	Enhanced green fluorescent protein (eGFP) *Oryza sativa* plastid ribosomal RNA operon promoter (*Prrn*) and *psbA* 5′-UTR	[202]
*Dunaliella tertiolecta*	Chlorophyceae, Chlamydomonadales, Dunaliellaceae	Halophilic	Biolistics *ereB* gene (erythromycin esterase)—Erythromycin	Silent site between *psbB/psbH*	Hydrolytic enzymes: xylanase, α-galactosidase, phytase, phosphate anhydrolase, and β-mannanase Endogenous P*psbD* Endogenous *psbA* 3′-UTR	[161]
*Desmodesmus armatus*	Chlorophyceae, Sphaeropleales, Scenedesmaceae	Resistant to high light and cold temperatures	Biolistics Mutated *PsbA* gene (S264K) conferring atrazine resistance—Atrazine	Endogenous *psbA* locus	-	[203]
*Haematococcus pluvialis (lacustris)*	Chlorophyceae, Chlamydomonadales, Haematococcaceae	Natural producer of high-value compound astaxanthin	Biolistics *Aad1* gene (aminoglycoside adenyltransferase)—Spectinomycin	Silent site between *rrn16S/rrn23S*	Endogenous P*rbcL* Endogenous *rbcL* 3′-UTR	[204]
*Haematococcus pluvialis (lacustris)*	Chlorophyceae, Chlamydomonadales, Haematococcaceae	Natural producer of high-value compound astaxanthin	Biolistics *Aad1* gene–Spectinomycin	Silent site between *rrn16S/rrn23S*	Phytoene desaturase (*pds*) to enhance endogenous astaxanthin accumulation Endogenous P*psbA* Endogenous *rbcL* 3′-UTR	[205]
*Haematococcus pluvialis (lacustris)*	Chlorophyceae, Chlamydomonadales, Haematococcaceae	Natural producer of high-value compound astaxanthin	Biolistics *Bar* gene–Bialaphos	Silent site between *rrn16S/tRNA-I* and *tRNA-A/rrn23S*	Antimicrobial peptide piscidin-4 (*ant1*) Endogenous P*rbcL* Endogenous *psbA* 3′-UTR	[206]
*Chlorella vulgaris*	Trebouxiophyceae, Chlorellales, Chlorellaceace	Thermo- and high-light-tolerant and high biomass producer	Biolistics *Aad1* gene—Spectinomycin	Silent site between *rrn16S/tRNA-I* and *tRNA-A/rrn23S*	Antimicrobial peptide NZ2114 and piscidin-4 (*ant1, ant2*) Endogenous P*16S* Endogenous *rbcL* 3′-UTR (*Aad1)* Endogenous P*rbcL* Endogenous *psbA* 3′-UTR (*ant1, ant2*)	[211]
*Parachlorella kessleri*	Trebouxiophyceae, Chlorellales, Chlorellaceace	Thermo- and high-light-tolerant and high biomass producer	Biolistics *Aad1* gene—Spectinomycin	Silent site between *rrn16S/tRNA-I* and *tRNA-A/rrn23S*	Endogenous P*psbA* Endogenous *psbA* 3′-UTR	[213]
*Picochlorum renovo*	Trebouxiophyceae, Chlorellales, Incertae sedis	Halophilic, thermotolerant, and high biomass producer	Biolistics *ereB* gene—Erythromycin	Silent site between *rrn16S/tRNA-I* and *tRNA-A/rrn23S*	Super folder GFP (sfGFP) Endogenous P*16S* Endogenous *16S* 3′-UTR	[173]
*Picochlorum renovo* and *celeri*	Trebouxiophyceae, Chlorellales, Incertae sedis	Halophilic, thermotolerant, and high biomass producer	Biolistics Chloroplast optimized *ptxD* isoform [77]—Phosphite	Silent site between *rrn16S/tRNA-I* and *tRNA-A/rrn23S*	PTXD Endogenous P*16S* Synthetic terminator	[78]
**Stramenopiles**						
*Nannochloropsis oceanica*	Eustigmatophyceae, Eustigmatales, Monodopsidaceae	Halophilic, high-lipid-accumulating	Electroporation *Sh ble* gene—Zeocyn	Endogenous *chlL* locus	Green Fluorescent protein (GFP) Endogenous P*rbcL* Endogenous *psbA* 3′-UTR	[48]
*Nannochloropsis gaditana*	Eustigmatophyceae, Eustigmatales, Monodopsidaceae	Halophilic, high-lipid-accumulating	Biolistics *bar* gene—Bialaphos	Silent site between *rrn16S/tRNA-I* and *tRNA-A/rrn23S*	Antimicrobial peptides ant1, ant2 Endogenous P*psbA* Endogenous *rbcL* 3′-UTR (*bar)* Endogenous P*rbcL* Endogenous *psbA* 3′-UTR (*ant1, ant2*)	[214]
*Phaeodactylum tricornutum*	Bacillariophyae, Bacillariales, Phaeodactilaceae	Halophilic, high-lipid-accumulating	Electroporation *cat* gene (chloramphenicol acetyltransferase)—chloramphenicol	Silent site between *rrn16S/tRNA-I* and *tRNA-A/rrn23S*	Enhanced green fluorescent protein (eGFP) Endogenous P*rbcL* Endogenous *rbcS* 3′-UTR	[215]

## 8. What Next?

Over the last two decades, microalgal research has significantly progressed, and today synthetic biologists have access to a profound knowledge and sophisticated tools to manipulate microalgal plastomes. Since its latest update [219], the chloroplast engineering toolbox has incorporated novel strategies, enabling the development of robust plastid-based expression platforms with negligible health and environmental risks. A major innovation in this field is the existence of alternatives to antibiotics for creating new strains, such as the PTXD/Phi platform [77], and the implementation of programmable/inducible systems, such as the temperature-controlled expression of recombinant products [110].

Undoubtedly, the greatest challenge lying ahead is the ability to perform complex genetic manipulations on algal chloroplasts, including the design of reduced synthetic chromosomes harboring the essential genetic information, along with transgenic elements. An early approach in this direction described the ex vivo assembly in a yeast–bacterial system of a hybrid algal plastome, containing genes from *Scenedesmus obliquus*, and its insertion into a non-photosynthetic *C. reinhardtii* strain [220]. Although full complementation could not be obtained in the resulting strain, this work paves the way for whole-plastome engineering in microalgae, and serves as a conceptual basis for further improvements.

Leveraging on recent advancements of plant biotechnology, it should be possible to stably introduce episomal vectors that replicate independently from the native plastome ((8) in Figure 1). This strategy was pioneered in tobacco and potato (*Solanum tuberosum*) plants, with the expression of transgenic cassettes encoded by synthetic mini-chromosomes harboring either a viral [221] or a chloroplast origin of replication [222]. In both cases, episomal expression was stable through generations, as well as after removal of the selective pressure, suggesting that the foreign system was stably maintained by the endogenous plastid DNA replication machinery. This innovative genetic engineering approach is also expected to find immediate applications in microalgae, especially in non-model species that are recalcitrant or not yet amenable to foreign DNA integration in their plastome.

In conclusion, due to these technological perspectives, a bright future awaits microalgal biotechnology, including the commercialization of their derived products. However, the full realization of this goal relies on two important aspects: market acceptance [223] and the enhancement of productivity [224] to withstand competition from traditional heterologous expression systems.

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
