# Peer review of "Harnessing the Algal Chloroplast for Heterologous Protein Production"

_microorganisms, 2022, doi:10.3390/microorganisms10040743_

Round 1

Reviewer 1 Report

The manuscript “Harnessing the algal chloroplast for heterologous protein production” submitted to microorganisms, gives a very comprehensive overview on the state-of the art experimental procedures and tools available to introduce and express protein coding genes in microalga plastids, mainly using Chlamydomomas reinhardtii as the recipient species of these technologies.  It is a very updated review of the potential of microalgal transplantomic biotechnology. The manuscript is well written and structured. I consider the review suitable for publication at microorganisms. I just have a few minor recommendations and corrections as detailed below:

- Throughout the manuscript, scientific names of species and the names of plastid and nuclear genes are not in italics. This needs to be corrected so that the article can be finally accepted for publication.

- Table 1 is very extensive, so I would suggest moving it to the supplementary material.

- Line 134. Full name for SMs.

- Line 228. Explain mixotrophic growth.

- Line 39. Explain the family of proteins to which Nac2 belongs in more detail.

- Line 167-169. I would appreciate to explain a bit more clearly involvement of the PTXDT gene as a selective marker and how, introduced in the nucleus of C. reinhardtii, can prevent culture infection by opportunistic parasites when algae are cultivated in non-sterile conditions.

Author Response

Dear Reviewer,

thank you for your comments. Please find below the datailed replies to your suggestions:

1) Throughout the manuscript, scientific names of species and the names of plastid and nuclear genes are not in italics. This needs to be corrected so that the article can be finally accepted for publication.

R1) Thank you for this observation. We have corrected all these mistakes in the manuscript.

2) Table 1 is very extensive, so I would suggest moving it to the supplementary material.

R2) Although we acknowledge that the  Table1 is very extensive, we would rather keep it in the main text since it contains relevant information which complement the corresponding paragraph of the manuscript.

3) Line 134. Full name for SMs.

R3) Done.

4) Line 228. Explain mixotrophic growth.

R4) Done.

5) Line 39. Explain the family of proteins to which Nac2 belongs in more detail.

R5) We have briefly described the protein role.

6) Line 167-169. I would appreciate to explain a bit more clearly involvement of the PTXDT gene as a selective marker and how, introduced in the nucleus of C. reinhardtii, can prevent culture infection by opportunistic parasites when algae are cultivated in non-sterile conditions.

R6) We have expanded this part of the text to provide a better explanation of the use of the ptxD gene coupled with phosphite fertilization for preventing microalgal culture infection and as novel chloroplast selectable marker.

Reviewer 2 Report

Unicellular lgae are unique organisms for heterologous production of proteins and other natural chemicals, because of their photoautotrophic growth. In this manuscript, the authors provide a comprehensive review of both background and current progress in the manipulation of algal chloroplasts. The manuscript is nicely organized and well written. I have only a few minor comments.

  1. The gene and species names should be italic. Please check and correct throughout the text.
  2. Also, please check and correct all letters and numbers that should be writtent in either superscript or subscript. Such as CO2 in line 40, PO33- in line 165, etc.
  3. Line 132, aadA1 is not aminoglycoside phosphotransferase. It should be aminoglycoside adenyltransferase.

Author Response

Dear Reviewer,

thank you for your comments. We have changed the text according to your suggestions.

1)The gene and species names should be italic. Please check and correct throughout the text.

R1) We have corrected all these mistakes in the manuscript.

2)Also, please check and correct all letters and numbers that should be writtent in either superscript or subscript. Such as CO2 in line 40, PO33- in line 165, etc.

R2) Done.

3)Line 132, aadA1 is not aminoglycoside phosphotransferase. It should be aminoglycoside adenyltransferase.

R3) We have corrected this mistake.